# Applications of Gas Sensing in Food Quality Detection: A Review

**DOI:** 10.3390/foods12213966

**Published:** 2023-10-30

**Authors:** Minzhen Ma, Xinting Yang, Xiaoguo Ying, Ce Shi, Zhixin Jia, Boce Jia

**Affiliations:** 1Information Technology Research Center, Beijing Academy of Agricultural and Forestry Sciences, Beijing 100097, China; minmatsura@163.com (M.M.); yangxt@nercita.org.cn (X.Y.); jiayazhixinjia@gmail.com (Z.J.); bocejia@outlook.com (B.J.); 2College of Food and Pharmacy, Zhejiang Ocean University, Zhoushan 316004, China; 3Key Laboratory of Cold Chain Logistics Technology for Agro-Product, Ministry of Agriculture and Rural Affairs, Beijing 100097, China; 4National Engineering Research Center for Information Technology in Agriculture, Beijing Academy of Agricultural and Forestry Sciences, Beijing 100097, China; 5National Engineering Laboratory for Agri-Product Quality Traceability, Beijing Academy of Agricultural and Forestry Sciences, Beijing 100097, China; 6Department of Agriculture, Food and Environment (DAFE), Pisa University, Via del Borghetto, 80, 56124 Pisa, Italy; 7College of Food Science and Nutritional Engineering, China Agricultural University, Beijing 100083, China

**Keywords:** gas sensor, gas sensor arrays, VOCs, quality monitoring, freshness prediction

## Abstract

Food products often face the risk of spoilage during processing, storage, and transportation, necessitating the use of rapid and effective technologies for quality assessment. In recent years, gas sensors have gained prominence for their ability to swiftly and sensitively detect gases, making them valuable tools for food quality evaluation. The various gas sensor types, such as metal oxide (MOX), metal oxide semiconductor (MOS) gas sensors, surface acoustic wave (SAW) sensors, colorimetric sensors, and electrochemical sensors, each offer distinct advantages. They hold significant potential for practical applications in food quality monitoring. This review comprehensively covers the progress in gas sensor technology for food quality assessment, outlining their advantages, features, and principles. It also summarizes their applications in detecting volatile gases during the deterioration of aquatic products, meat products, fruit, and vegetables over the past decade. Furthermore, the integration of data analytics and artificial intelligence into gas sensor arrays is discussed, enhancing their adaptability and reliability in diverse food environments and improving food quality assessment efficiency. In conclusion, this paper addresses the multifaceted challenges faced by rapid gas sensor-based food quality detection technologies and suggests potential interdisciplinary solutions and directions.

## 1. Introduction

Olfaction is one of the most useful systemic senses in humans. In mammals, the perception of odors relies on olfactory nerve receptors. These receptors are stimulated to produce signals, which are then transmitted via nerves to the brain, enabling the perception of different food odors [1]. Food quality control is of paramount importance as it is directly related to human health and well-being. Traditional food quality testing methods, typically encompassing sensory assessment, physical assessment, chemical assessment, and microbiological assessment, continue to be the prevailing approaches for evaluating food quality [2]. Despite their widespread use, these conventional testing methods are known for being tedious and time-consuming. Gas chromatography mass spectrometry (GC-MS) is a powerful and sophisticated analytical instrument capable of accurately analyzing gas compositions. It works by separating gas mixtures into individual components through gas chromatography and then identifying and quantifying these components using mass spectrometry. However, this analytical method is costly and time-consuming, and requires a relatively bulky analyzer in a specialized laboratory environment with experienced operators. It lacks real-time monitoring capability for target substances, and exhibits poor availability in the field. Similarly, traditional sensory assessment is vulnerable to human subjectivity, variations in expertise, and other objective factors such as the environment. As a result of technological advancements, the food industry is increasingly interested in rapid and precise identification, quantification, and monitoring of volatile organic compounds (VOCs) in food products. These VOCs encompass a wide range of compounds, including organic acids, esters, aliphatic alcohols, polyphenols, aldehydes, ketones, amino acids, and long-chain and naphthenic hydrocarbons. In response to this growing demand, there is a need to develop simple devices that meet the critical requirements for cost-effectiveness, ease of use, and real-time monitoring. These devices are expected to find applications in various areas of the food industry, facilitating quality control, freshness assessment, and safety monitoring. Gas sensors function similarly to the human nose, allowing them to detect specific gases in a given area or continuously measure gas composition and concentration. By combining multiple non-specific gas sensors into sensor arrays, it becomes possible to determine the quality or identify the type of a compound. As a result, gas sensors have become a promising method for assessing food quality, facilitating the monitoring of food product quality and freshness. Over the past decades, significant advancements have been achieved in the development and application of gas sensors, employing diverse sensing materials and transduction platforms. Gas sensors constitute the core of electronic nose systems, primarily consisting of a gas sensor array, a signal preprocessing module, and a pattern recognition engine [3,4]. These gas sensors offer several advantages, including short response times, rapid detection, a wide detection and evaluation range, and good repeatability [5]. By employing gas sensors with distinct properties, a sensor matrix can be formed, enabling the simultaneous monitoring and identification of multiple gases in the environment. This capability positions gas sensor arrays among the most advanced monitoring instruments in the world.

This paper provides a comprehensive review of recent advances in gas sensor technologies relevant to food quality monitoring. It covers various types of gas sensors, including metal oxide (MOX) sensors, colorimetric sensors, electrochemical sensors, surface acoustic wave (SAW) gas sensors, and sensor arrays. We have analyzed the main features and working principles of each gas sensor, along with their research progress and application in food quality monitoring. This comprehensive analysis will assist researchers and experts in selecting the most suitable food sensor for their specific detection requirements. Furthermore, we discuss the potential of gas sensors in monitoring food quality and freshness, presenting insightful perspectives. Finally, the paper concludes with recommendations for future research directions in this promising field.

## 2. Overview of Gas Sensor Technology on Food Quality Monitoring

Over the past few decades, significant advancements have been made in gas sensor technology, resulting in the development of various sensor types based on different operating principles. In Figure 1, a flowchart of a gas sensor-based food gas detection system is presented. These sensors are classified into distinct categories, including MOX gas sensors, SAW gas sensors, colorimetric gas sensors, and electrochemical gas sensors. Researchers are continuously engaged in the exploration of sensor materials with enhanced intrinsic properties, seeking materials with improved rigidity or flexibility. In current research endeavors, catalysts and other elements are introduced into sensor materials to optimize processing techniques. This optimization aims to enhance crucial parameters, including the sensitivity, response time, and cross-sensitivity of gas sensors [6].

We conducted a comprehensive review of articles related to gas sensors and their application in monitoring food quality from January 2011 to June 2023 on the Web of Science platform. Our search was narrowed down to articles with the keywords ‘gas sensors’ and ‘food quality’ to ensure precision. This thorough screening process resulted in the identification of 154 published articles, all of which specifically focused on the innovative use of gas sensors for monitoring and preserving food freshness. Among these articles, 113 were research articles, accounting for 73.34% of the total. Figure 2A presents an overview of the published articles on food quality monitoring by gas sensors during the specified period, showcasing a clear and consistent increasing trend. Additionally, Figure 2B displays the distribution of gas sensor categories used in the published articles for monitoring food quality from January 2011 to June 2023. Gas sensors are categorized based on differences in operating principles and characteristics, with both MOX gas sensors and colorimetric sensors constituting the majority, each contributing 36% of the total. Following closely are electrochemical gas sensors (7%) and SAW gas sensors (4%). The remaining 17% is attributed to other gas sensors. These findings shed light on the prevalent use and significance of metal oxide and colorimetric sensors in gas sensor-based food freshness monitoring research.

## 3. Gas sensors for Food Quality Monitoring

### 3.1. Metal Oxide Sensors

MOX sensors have earned widespread application in monitoring water quality, food quality, chronic diseases, and industrial processes, owing to their distinctive advantages in electrical, electrochemical, and biocompatible properties [7]. These unique attributes guarantee their efficacy in sensor manufacturing. The fundamental functionality of an MOX sensor relies on reactions occurring when a gas interacts with a metal oxide or metal oxide semiconductor material adsorbed on its surface. These interactions induce changes in the sensor material’s conductivity, voltammetric properties, or surface potential. A key characteristic of MOX sensors during operation is the carrier motion, governed by the semiconductor properties of the material. This results in MOX sensors exhibiting distinct responses to various target gases [8]. Initially, oxygen atoms adsorb on the MOX surface, extracting electrons from the n-type material to form anionic oxygen entities such as O_2_^−^, O^2−^, O^−^, etc., which reduce the carrier concentration on the MO surface, leading to a change in resistance [9]. Gas sensors utilizing metal oxide semiconductor materials can be classified into two primary types, distinguished by the type of carriers they employ: n-type sensors, where electrons act as the primary carriers, and p-type sensors, where holes are the primary carriers. Examples of n-type metal oxide semiconductors encompass zinc oxide, tin dioxide, titanium dioxide, and iron (III) oxide. Conversely, p-type metal oxide semiconductors include nickel oxide and cobalt oxide, among others [10]. Figure 3 presents a schematic diagram illustrating the structure of a metal oxide semiconductor gas sensor. Traditionally, gas sensors have relied on materials such as SnO, SnO_2_, and Fe_2_O_3_. However, recent advancements in research and development have led to the exploration of novel materials, including single metal oxide materials, composite metal oxide materials, and mixed metal oxide materials. These endeavors aim to elevate sensor performance and expand the application possibilities of gas sensors.

MOS gas sensors can also be classified into two types based on their gas-sensitive mechanisms: resistive semiconductor gas sensors and non-resistive semiconductor gas sensors. These classifications are determined by the specific ways in which the sensors interact with and respond to target gases [11]. Resistive semiconductor gas sensors primarily function as impedance devices and are mainly composed of metal oxide ceramic materials. These sensors utilize metal oxide films, such as SnO_2_, ZnO, Fe_2_O_3_, and TiO_2_, among others. The adsorption of gas molecules on the semiconductor surface causes the formation of a surface state that affects the conductivity of the semiconductor material, which in turn changes the electrical resistance [12]. Resistive metal–semiconductor gas sensors offer numerous advantages that make them highly desirable for various applications. These sensors are not only affordable but also relatively easy to fabricate, allowing for mass production. One of their key strengths is their high sensitivity, enabling precise detection of target gases. Moreover, they exhibit rapid response times, ensuring real-time monitoring capabilities. These sensors have a long lifespan when properly maintained, making them durable for extended use. However, water molecules in humid environments can adsorb to the surface of the sensor, leading to the formation of surface states, which can affect the conductivity of the sensor. There are a number of resistive semiconductor gas sensors that have taken moisture protection measures, such as moisture-proof packaging or special coatings, to minimize the effects of humidity. Furthermore, their straightforward circuitry requirements simplify integration into systems, enhancing their overall usability [13]. Non-resistive semiconductor sensors, on the other hand, utilize other properties of the semiconductor material to detect gases, such as capacitance, inductance, and the Hall effect [12]. Non-resistive semiconductor gas sensors, while useful for detecting certain gases like hydrogen, do have some drawbacks. These include elevated operating temperatures, limited selectivity towards the desired gas, variability in component parameters, less-than-optimal stability, high power demands, and susceptibility to poisoning when exposed to a mixture of sulfide gases. For the detection of combustible gases like hydrogen, non-resistive semiconductor gas sensors primarily rely on monitoring the changes in the current or voltage of the gas-sensitive element in response to gas concentration. To optimize sensitivity and selectivity, adjustments can be made to the catalytic metal type, film thickness, and operating temperature. It is worth noting, however, that the fabrication process of non-resistive metal-semiconductor gas sensors is more intricate and costly compared to their resistive counterparts.

The response characteristics of metal oxide sensors to gases, including the gas response rate and gas selectivity, are primarily influenced by various factors. These factors encompass the surface area of the sensing material, receptor density, agglomeration, porosity, acid-base properties, catalyst, and ambient temperature of the metal oxide sensor. Each of these parameters plays a critical role in shaping the performance and sensitivity of the sensor in detecting and distinguishing different gases. A larger surface area enhances gas interaction, leading to improved sensor response, while the density of receptors on the sensor’s surface affects its ability to detect and respond to specific gases. The degree of agglomeration in the sensing material can impact gas diffusion and alter the sensor’s response, and highly porous materials can facilitate gas diffusion, influencing the sensor’s sensitivity. The chemical properties of the metal oxide sensor’s surface influence its interactions with different gases, and the presence of catalysts can modify gas reactions, affecting the sensor’s selectivity. Furthermore, changes in ambient temperature can influence the sensor’s response and sensitivity to gases. Understanding and optimizing these parameters are crucial for maximizing the efficiency and accuracy of metal oxide sensors in gas detection applications [14]. The addition of catalysts to the material of MOX sensors can significantly enhance their sensitivity to specific gases. Nevertheless, it is essential to consider that an excessive amount of catalyst may potentially lead to a reduction in the response rate of the metal oxide sensor. Therefore, achieving an optimal balance between the catalyst concentration and response rate becomes crucial for ensuring the efficient and accurate detection of target gases [15]. Furthermore, the response and selectivity of MOX sensors to specific gases are significantly influenced by their grain size. Smaller grain sizes tend to exhibit higher sensitivity compared to larger ones. Moreover, the resistance of MOX sensors is significantly affected by temperature variations. Temperature changes can impact the sensors’ performance and accuracy in gas detection. Therefore, it is essential to carefully consider and control temperature conditions to ensure reliable and precise gas detection using metal oxide sensors [16]. In a typical initial environment, the resistance of MOX gas sensors is generally relatively high. However, when exposed to a heat source and/or specific relevant gases, the resistance of MOX gas sensors undergoes changes. The direction (increase or decrease) and magnitude of resistance change depend on factors such as the sensor’s material, operating temperature, type of target gas, and concentration of the target gas. Once the heat source dissipates and/or the test gas is no longer present, the resistance of the metal oxide sensor returns to its initial level.

Abundant research has demonstrated the efficacy of metal oxide sensors in various applications within the realm of food quality detection, with successful use in monitoring food safety and spoilage [17]. The focus of research on food freshness monitoring based on metal oxide sensors remains centered on aquatic and livestock meat, as well as fruit and vegetables. For instance, the installation of arrays consisting of metal oxide gas sensors in refrigerators allows for intelligent monitoring of food products’ freshness and preservation management. In food processing and production [18,19], such as refrigeration, pasteurization, and vacuum packaging, metal oxide sensors play a pivotal role in enabling intelligent monitoring. Researchers have utilized metal oxide sensors to monitor the presence of total VOCs in aquatic products, chicken, and food packaging [20,21,22]. Under various environmental conditions, metal oxide sensors hold excellent potential for wireless sensing and online monitoring applications due to their high sensitivity, broad spectrum, relatively low cost, and durability. As research proceeds, the continuous improvement of the selectivity, sensitivity, and stability of metal oxide sensors remains a major research area, and it is essential for enhancing the accuracy and reliability of food quality monitoring [2].

### 3.2. Surface Acoustic Wave Gas Sensors

The SAW gas sensor operates by utilizing a SAW delay line or resonator device, which is connected to the feedback loop of a high-frequency amplifier. The presence of a polymer film on the piezoelectric material allows for the adsorption of VOCs. This adsorption, combined with the phase compensation network, results in a covariance within the SAW oscillator [5,23]. Figure 4 displays a schematic diagram of the structure of SAW gas sensors. IDTs (interdigital transducers) are key components of the sensors used to generate and receive surface acoustic waves. The basic principle of SAW sensors involves the adsorption of a sensitive membrane material on the sensor’s surface to a specific gas being measured. This adsorption leads to fluctuations in the speed of acoustic surface wave propagation, resulting in a change in the frequency of oscillation. As a result, the gas to be measured is detected [2]. Surface acoustic wave (SAW) gas sensors have been widely used in chemical and biosensing applications [24,25].

SAW gas sensors offer several advantages, including high selectivity, high sensitivity, stability across a broad temperature range, minimal response to humidity, and excellent repeatability [26]. These features contribute to the effectiveness and reliability of SAW gas sensors in detecting and monitoring specific gases with precision and consistency. Moreover, the utilization of integrated circuits in the plane process enables SAW sensors to achieve a single, multi-functional, and integrated design, resulting in reduced sensor size and weight. This miniaturization and integration facilitate ease of use and portability, ultimately reducing measurement costs [25]. Researchers have successfully employed SAW gas sensor-based food monitoring technology to identify and distinguish various types of produce, evaluate flavor characteristics [27], and determine the shelf life of food products [28].

The main areas of research for SAW sensors encompass the advancement of new sensor technologies, the development of materials with excellent electroacoustic properties for fabricating high-frequency devices on silicon, and the enhancement of sensor sensitivity and selectivity. All these efforts are undertaken while striving to reduce manufacturing costs. These research directions aim to propel the continuous improvement and innovation of SAW sensor technology, enabling its broader application in various industries and fields [29]. In the field of coating biomaterials, the widespread utilization of polymers has led to continuous advancements in the development of new polymers and coating techniques. A novel approach involves the application of biomolecules such as DNA, peptides, and proteins for sensing purposes in gas detection. This approach introduces new interaction mechanisms, enhancing selectivity as well as other desirable physical properties, thereby contributing to the overall improvement of gas sensor performance [30].

### 3.3. Colorimetric Sensors

The colorimetric sensor, categorized as one of the optical gas sensors, is designed to mimic the olfactory system of mammals, producing a unique and complex response to each specific gas. This enables automatic photoelectric colorimetric measurements, making it suitable for the analysis of volatile organic gases, amines, and more in the field of food monitoring [31]. Although colorimetry is an old, simple, and fast analytical technique that can be quantified directly with the naked eye, it has been given new possibilities through modern digital imaging methods. The colorimetric sensor functions according to the Beer–Lambert law, which arises from the chemical interaction between the analyte molecules and the sensor’s active center. This interaction goes beyond mere physical adsorption and triggers a colorimetric change. Colorimetric sensor arrays leverage digital imaging to offer a straightforward, effective, and highly sensitive approach for the rapid detection and identification of various chemical substrates [32]. Furthermore, colorimetric sensor arrays based on chemically responsive dyes possess numerous advantages, including excellent selectivity, high sensitivity, non-destructive analysis, cost-effectiveness, low detection limit, and rapid response time. As a result, this technology holds significant potential for enabling “odor visualization” of food products, offering a valuable tool for assessing and monitoring their freshness and quality [33].

Colorimetric sensors can be classified into two categories: chemical sensors and biological sensors, based on the type of interaction involving chemical or biological molecules [34]. In colorimetric sensor arrays, the main sensing units are chemo-responsive dyes, which exhibit color changes in response to specific chemical environments [33]. This characteristic enables the detection and analysis of various substances by monitoring the corresponding colorimetric changes in the sensor array. Typically, chemically reactive dyes are selected based on their high absorption coefficients and/or emission quantum yields in solution and their ability to respond to specific target gases. These dye molecules are then immobilized onto a solid support using methods such as adsorption, entrapment, ion exchange, or covalent binding. For instance, a novel colorimetric sensor array based on a nanoporous titanium dioxide membrane has been developed for detecting low concentrations of trimethylamine in meat [35]. In another study, manganese tetraphenylporphyrin (MnTPP) was employed as a sensitive material in a homogeneous optical waveguide sensor system to detect trimethylamine [36]. These examples illustrate the diverse applications of colorimetric sensor arrays in the detection and analysis of specific substances, offering a promising alternative for assessing the freshness of fish and seafood. Furthermore, the performance of a colorimetric sensor is significantly influenced by the selection of a suitable solid support and the immobilization method. These factors impact its selectivity, sensitivity, dynamic range, calibration, response time, and stability. In the case of biological molecules, nanoparticle-based sensors have garnered significant interest due to the unique properties exhibited by nanomaterials, such as biocompatibility, electrical conductivity, and catalytic activity. These characteristics make nanoparticle-based sensors highly suitable for various biological sensing applications [37]. Biosensors have been developed that are capable of detecting biological macromolecules such as antigens, antibodies, proteins, and DNA, as well as facilitating interactions, enzyme detection, and microbial recognition. In recent years, the use of gold and silver nanoparticles (AuNPs and AgNPs) as colorimetric sensors has gained significant attention. These nanoparticles exhibit exceptional sensitivity to alterations in the surrounding environment, thanks to their high surface-to-volume ratios and distinctive optical properties [37,38]. The unique characteristics of metal nanoparticles, such as gold, silver, copper, and platinum, including surface plasmon resonance (SPR), make them valuable for the highly sensitive detection of biomolecules. Their SPR properties enable the detection and sensing of biomolecular interactions with exceptional sensitivity. 

### 3.4. Electrochemical Gas Sensors

Electrochemical gas sensors are designed to measure gas concentration by undergoing a chemical reaction with the target gas, resulting in the generation of an electrical signal that is proportional to the gas concentration [39]. Figure 5 illustrates the structure of an electrochemical gas sensor. The CE (counter electrode) is used to maintain charge balance between the electrodes along with the WE (working electrode), which provides the current path and maintains the stability of the cell. The WE is the electrode used to detect and measure the target analyte (usually a gas). When the target analyte reacts electrochemically with the working electrode, it generates a current or voltage signal that can be used to measure the concentration of the analyte or other relevant parameters. The WE is usually selective so that it responds only to a specific analyte. The RE (reference electrode) is used to provide a reference point of known potential [40]. This helps to accurately measure the change in current or voltage during an electrochemical reaction and compare it to the known potential to derive information about the concentration of the analyte. The RE is usually made using saturated salt solution or other special electrode materials. The electrode and electrolyte of an electrochemical gas sensor come into contact with the surrounding air and are enclosed by a porous membrane. Typically, mineral acids are used as the electrolyte, although some sensors may employ organic electrolytes. The electrode is usually housed within a plastic enclosure, featuring gas inlet holes and electrical contacts. As the gas diffuses through the back of the porous membrane, it reaches the working electrode of the sensor. Here, the gas undergoes oxidation or reduction, resulting in an electrochemical reaction that generates a current in the external circuit. A resistor is connected between the electrodes and serves to promote the flow of the current, which is proportional to the concentration of the gas being measured. In addition, electrochemical sensors can be categorized into several types depending on the principle of operation, including amperometric sensors, potentiometric sensors, impedance sensors, photoelectrochemical sensors, and electrogenerated chemiluminescent sensors [41,42]. 

Electrochemical sensors possess unique and crucial attributes that are intricately tied to the properties of the electrode material. Customization becomes imperative to ensure that these sensors align precisely with their intended applications and the specific measurement principles governing their operation. Pereira et al. [15] proposed an anthocyanin-sensitive gelatin–zinc oxide nanocomposite film for meat quality assessment. Ma et al. [43] developed a low-energy monitoring system using electrochemical sensors to real-time monitor the concentration of ethylene gas released from apples, pears, and kiwis. Potentiometric sensors have also demonstrated wide utility in fish quality monitoring [44,45]. Additionally, nanomaterials, such as metal nanoparticles, metal oxide nanoparticles, and carbon-based materials, have been investigated as electrode modifiers or new electrode materials. A detailed description of nanomaterial-based electrochemical sensors is given by Khaled et al. [46] Nanomaterials (NMs) offer several advantages, including a large surface area, good catalytic activity, easy synthesis routes, and favorable optical, physical, electrical, and mechanical properties. Moreover, NMs can serve as catalysts for electrochemical reactions, reducing the energy required for these reactions and facilitating their reversibility. Overall, electrochemical gas sensors offer the advantages of high selectivity, high sensitivity, fast response, and long life, but compared to MOX gas sensors, electrochemical sensors require regular maintenance and calibration to ensure accuracy and stability, and their performance is limited under extreme temperature conditions. In multi-gas environments, gas cross-talk can occur, affecting accuracy.

### 3.5. Sensor Arrays

Gas sensor arrays mimic the complex network of olfactory receptors in the human olfactory system. They are systems of multiple gas sensors used to detect, analyze, and identify the composition and concentration of different gas mixtures. The main advantages of gas sensor arrays are their high sensitivity and high selectivity, making them valuable tools for gas composition and concentration analysis [47]. The design of the sensor array plays a crucial role in achieving high sensitivity and selectivity, and thus significantly impacts the overall performance of the gas monitoring system. By combining the response signals of multiple sensors, more accurate and reliable gas identification results can be obtained. Additionally, different chemically sensitive materials have varying response characteristics to the gas being detected, enabling selective identification and analysis of gas mixtures. Lotfivand et al. [48] proposed a new structure with strong discrimination in detecting complex odors, excellent performance in case of sensor failure, and a comparison with general structures. Wang et al. [49] developed a sensor array based on MOS gas sensors for easy, direct, and real-time assessment of food freshness by monitoring odor changes in a refrigerator. Zhen et al. [50] proposed an integrated sensor array consisting of multiple silicon-based chemically sensitive field-effect transistors (CSFETS) that can simultaneously and sensitively measure the signature gases of food spoilage. Moreover, a comprehensive classification of the practical applications of gas sensor arrays has been summarized in a paper by Wang et al. [5].

Consequently, gas sensor arrays have emerged as a promising solution to overcome the limitations associated with single sensors that are solely sensitive to particular gases. In addition to excellent sensitivity and selectivity, because each sensor in the sensor array can respond independently to its target gas, parallel processing of information allows for a faster response, and redundancy and diversity in the array allows for increased reliability and speed of detection by cross-referencing data from multiple sensors. Therefore it is widely used in various fields. Moreover, when combined with advanced data analysis and pattern recognition methodologies, gas sensor arrays can enable automated identification, classification, and monitoring of intricate gas mixtures within the environment.

## 4. The Application of Gas Sensors in Food Quality Monitoring

Food undergoes continuous deterioration during processing, storage, and transportation, leading to a decline in its quality. Volatile compounds, which are crucial indicators of freshness, experience significant changes throughout food processing and storage, making gas sensors essential for detecting them and assessing food quality [51]. Figure 2C illustrates that gas sensor-based freshness monitoring studies have primarily focused on aquatic products, meat products, and fruit. While other food types have also been included, there are fewer studies on specific food products such as edible oils, snack foods, dairy products, egg products, and cereal products. For a comprehensive overview of gas sensor applications in monitoring different types of food, Table 1 provides a classification of food types based on gas sensor applications for freshness monitoring, after careful screening.

### 4.1. Aquatic Products

Aquatic products serve as a broad categorization encompassing both aquatic animal products and plant products, as well as their processed counterparts, derived from both marine and freshwater fisheries. This includes fish, shrimp, crab, shellfish, algae, sea animals, and other fresh products obtained through fishing and aquaculture. Additionally, it encompasses processed products that have undergone various treatments such as freezing, curing, drying, smoking, cooking, canning, and integration. Aquatic products play a significant role in daily human food production and consumption [76]. Throughout the process of spoilage, the quality of fish products progressively diminishes, leading to the release of distinct volatile compounds such as trimethylamine, dimethylamine, and biogenic amines [41]. Presently, gas sensors are employed to monitor the freshness of aquatic products by detecting the presence of these volatile compounds. Among the research articles related to aquatic product quality detection from January 2011 to June 2023, 11 articles were based on MOX gas sensors, with a focus on the period until 2022, while 15 articles were based on colorimetric sensors, with most of them concentrated after 2022.

In the field of MOX sensors for the detection of fish quality, Kawabe et al. [77] conducted a study to explore the applicability and reliability of metal oxide sensors for monitoring volatiles in live oysters. In this study, various aldehydes, alcohols, and carboxylic acids, along with DMS and TMA, were selected as detectors. Through principal component analysis of the gas sensor data and sensory evaluation, the researchers found that the odor of live oysters exposed to air deteriorated during storage. Notably, the study revealed that most aldehydes decreased, while trimethylamine and volatile carboxylic acids accumulated in deteriorating live oysters stored under air exposure conditions. Consequently, trimethylamine and volatile carboxylic acids emerged as useful indicators of oyster freshness, suggesting the potential application of gas sensors in oyster freshness monitoring. In another study [20], a sensing system consisting of MOX sensors was employed to monitor the freshness of salmon, proving to be equally effective. In this research, fish spoilage was modeled using an artificial intelligence algorithm to monitor changes in TVC and TVB-N of the samples during 15 days of storage, achieving a remarkable correctness rate of 96.87%. Additionally, Han et al. [52] utilized nine MOX gas sensors to monitor the freshness of rhubarb fish stored at 4 °C for different durations. The results demonstrated that a single gas sensor system was sufficient to classify and assess the freshness of samples stored for different periods at 4 °C. Furthermore, a combination of two sensors achieved an even higher resolution.

In the field of colorimetric sensors for fish quality detection, Lv et al. [78] employed a colorimetric sensor array to monitor the freshness of horse mackerel, with a specific focus on analyzing TMA as a characteristic gas. The experimental findings demonstrated the effective use of the colorimetric sensor array for monitoring the freshness of mackerel. Furthermore, the study explored the incorporation of metalloporphyrin into the material construction of the colorimetric sensor array. In recent years, there has been significant attention on colorimetric sensors based on anthocyanins, specifically targeting volatile amines as the characteristic gases. Researchers such as Zhai et al. [79], Milad et al. [80,81], Kamer et al. [82], and Zheng et al. [83] have successfully utilized anthocyanins as raw materials to develop detection labels or colorimetric films. These studies selected volatile amines as characteristic gases and validated the effectiveness of anthocyanin-based detection labels or colorimetric films for monitoring the freshness of aquatic products. Additionally, fluorescent label-based colorimetric sensors have also been employed for freshness testing of aquatic products [84].

Furthermore, gas sensors have been extensively studied for monitoring and predicting the freshness and quality of fish products stored at various temperatures, employing computer-aided modeling techniques. This approach allows for continuous monitoring and assessment of fish product quality based on data collected by gas sensors. Hui et al. [85] introduced a gas sensor-based model for predicting the freshness of grass carp stored at 4 °C, achieving a prediction accuracy of 87.5%. Li et al. [53] developed an electronic nose system comprising metal oxide sensors and established a prediction model for the K value of large yellow croaker through linear fitting regression of the K value and maximum signal-to-noise ratio value. The model exhibited a high regression coefficient (R^2^) of 0.96 and a prediction accuracy of 83%, further supported by a regression coefficient of 0.83. In a different study [86], an electronic olfactory bionic system based on an array of colorimetric sensors was used to monitor the freshness of rhubarb. The researchers developed a support vector regression (SVR) model to investigate the correlation between the colorimetric sensor signals and the TVB-N and TVC values of the samples. The experimental findings demonstrated the colorimetric sensor array’s capability to qualitatively predict the freshness of unknown fish samples by monitoring TVB-N and TVC. Moreover, this non-destructive approach allowed for the effective assessment of fish freshness.

### 4.2. Meat Products

Meat products are an integral and essential part of the human diet, and their rich nutrient content makes them highly susceptible to microbial contamination during processing, storage, and transportation, leading to spoilage [87]. To combat this issue, gas sensors have been increasingly utilized to detect volatile compounds such as trimethylamine, volatile fatty acids, biogenic amines, alcohols, ammonia, and carbon dioxide, which are released during the spoilage of meat products. These gas sensors play a crucial role in monitoring and ensuring the freshness of meat products [88]. A comprehensive review of research articles related to the detection of meat product quality from January 2011 to June 2023 revealed seven articles based on MOX gas sensors, with a focus on the period until 2022. Additionally, 10 articles were centered around colorimetric sensors, and notably, six of these articles were published after 2022. By effectively employing gas sensors, the industry can enhance the quality control and shelf life management of meat products, ensuring their safety and freshness for consumption. Continuous research and advancements in gas sensor technology hold significant potential for further improving the monitoring and preservation of meat products in the future.

In the field of quality detection for meat products, metal oxide gas sensors have been utilized to monitor freshness changes in fresh chicken meat during storage. Edita et al. [21] investigated the application of metal oxide gas sensors to test the selection of volatile fatty acids, which are known to represent meat spoilage. The results from their research demonstrated the potential and effectiveness of using gas sensor systems to assess the freshness of fresh chicken. Similarly, Tang et al. [89] explored a rapid evaluation method for TVB-N contents in chicken meat using a gas sensor array. Their experiment employed a sensor array composed of metal oxide gas sensors to monitor the gas composition changes within fresh chicken meat stored at 4 °C. Additionally, the same sensor array was utilized to detect the TVB-N content in the chicken meat. Based on the gathered data, Tang et al. developed a prediction model for TVB-N in chicken meat, achieving an impressive prediction accuracy of 93.3%.

In the realm of food quality detection, colorimetric sensors have emerged as a promising approach. Huang et al. [35] developed a nanoporous colorimetric sensor array (NCSA) to effectively monitor TMA production, which is a critical indicator of meat spoilage. This sensor demonstrated the ability to visually detect TMA gas concentrations ranging from 10 ppm to 60 ppb, with a significant response. Similarly, anthocyanin-based colorimetric sensors have shown potential for detecting volatile amines and assessing the freshness of meat products. The studies by Kilic et al. [90], Wang et al. [90], and Zheng et al. [91] used different anthocyanins as raw materials to create detection labels with volatile amines as target gases and verified the validity of freshness monitoring of anthocyanin-based colorimetric sensors for pork, chicken, and other meat foods. Furthermore, Isabel et al. [58] innovatively combined a colorimetric sensor with pork packaging to assess freshness. The colorimetric sensor underwent a color change in response to increasing carbon dioxide levels caused by bacterial growth in the packaged meat. By capturing photographs of these color changes with a smartphone, the study successfully demonstrated a low-cost and rapid method for detecting pork freshness. The grayscale measurement of color information correlated well with bacterial growth and carbon dioxide gas released from the packaged meat, providing a promising new approach for detecting freshness.

Regarding the detection of food quality using electrochemical gas sensors, Wojnowski et al. [92] conducted a study to monitor the freshness changes in fresh chicken meat. They employed an electrochemical sensor and accurately predicted the biogenic amine index of chicken samples during refrigeration. The study’s results highlight the high precision and accuracy achieved by using an electrochemical gas sensor-based detection system to assess the freshness of chicken based on biogenic amine measurements. Additionally, the potential commercial viability of this approach is bolstered by its low cost.

### 4.3. Fruit Products

Fruit plays a vital role in the human diet, offering essential vitamins, minerals, and nutrients [93]. Being perishable, fresh fruit releases characteristic alkenes, esters, aldehydes, and alcohols during decay. For instance, the characteristic odor components of apples include substances like ethanol, isobutanol, and benzyl alcohol [94]. In recent years, gas sensors have gained increasing prominence in monitoring the quality and freshness of fruit, with studies from January 2011 to June 2023 primarily focusing on two types: metal oxide gas sensors and colorimetric sensors.

Sanaeifar et al. [60] conducted a study focused on freshness monitoring and prediction in bananas using metal oxide sensors. Ethylene—a characteristic gas released during fruit ripening—served as the key parameter, and the study established a correlation between the sensor array response and banana quality indicators. The results demonstrated the potential of metal oxide gas sensors for effectively monitoring and predicting the freshness of bananas. In a separate study, Beniwal et al. [61] employed a novel metal oxide gas sensor to monitor the freshness of apples. The sensor—a thin-film Ni-SnO_2_ sensor prepared via a simple and cost-effective sol–gel spin-coating technique—showed promising results in detecting ethylene released during apple spoilage. Furthermore, the sensor’s high sensitivity and selectivity were verified through experiments on decaying bananas and kiwis, as well as non-ripe fruits like oranges and grapes, demonstrating its effectiveness for monitoring decaying apples. 

Furthermore, Ma et al. [43] introduced a novel detection system utilizing an electrochemical gas sensor to monitor ethylene levels in fruit. The study investigated the system’s effectiveness in assessing the freshness of apples, pears, and kiwis. Before measuring the concentration of ethylene gas released by the fruit, the researchers established a calibration curve using standard ethylene gas. This curve demonstrated a linear relationship between the sensor’s response and ethylene gas concentration in the range of 0–10 ppm, achieving a high R^2^ value of 0.9976. These experimental findings underscore the system’s remarkable sensitivity, affordability, and compactness, validating its efficacy for monitoring fruit freshness.

### 4.4. Dairy Products

Spoilage of milk and its products primarily arises due to alterations in sensory characteristics resulting from protein decomposition, fat oxidation, and microbial metabolic activity. Dairy products, being nutrient-rich environments conducive to the growth and proliferation of spoilage and pathogenic microorganisms, pose an elevated risk of compromised quality and food safety [95]. To address quality and freshness concerns in dairy products, gas sensors have emerged as valuable tools, offering promising applications for real-time monitoring of the condition of dairy items. By detecting changes in gas emissions related to spoilage or microbial activity, these sensors ensure the optimal quality and safety of dairy products.

Rayappan et al. [66] employed a commercially available metal oxide semiconductor sensor array to monitor the quality of raw milk. Calibration of these sensors was conducted across a range of concentrations for VOCs commonly found in raw milk, such as ethanol, trimethylamine, acetaldehyde, dimethyl sulfide, and acetic acid. These VOCs are typically produced due to microbial contamination, chemical reactions, and genetic factors associated with cows. The study demonstrated the gas sensor’s selectivity and recognition efficiency in providing real-time identification of raw milk quality. In a separate investigation, Mohamed et al. [68] developed a colorimetric sensor using SiO_2_ nanoparticles and Schiff’s reagent to monitor the freshness of pasteurized whole milk stored at different temperatures (7, 13, 15, and 19 °C). The experimental findings indicated that the colorimetric sensor exhibited distinct color variations due to VOCs generated by spoilage bacteria at storage temperatures of 13, 15, and 19 °C, indicating microbial growth in the milk. Notably, the overall color difference (∆E) of the sensor showed a strong correlation (R^2^ = 0.81–0.96) with the aerobic plate count, a measure of microbial presence in the milk.

### 4.5. Grain and Oil Products

During the storage of grains, a combination of enzymatic activity, microbial presence, and storage pests leads to significant changes in their chemical composition and volatile components. Over time, these alterations cause grains to lose their original color, aroma, taste, nutritional composition, and overall food quality. Unfortunately, some storage conditions can even lead to the production of harmful substances such as aflatoxins, which can contaminate grains when certain fungi, especially Aspergillus, proliferate in warm, humid storage environments, posing serious risks to human health, including liver damage and an increased risk of liver cancer. 

Guan et al. [72] conducted a study where they monitored the freshness of rice using a colorimetric sensor array. The researchers initially analyzed the VOCs of rice samples at various storage times using GC-MS, and the results demonstrated the sensor’s capability for effective characterization. Moreover, the colorimetric sensor array successfully differentiated rice with storage periods exceeding 6 months, showcasing its ability to accurately classify rice based on different storage durations and monitor its quality.

In a similar vein, edible fats and oils undergo chemical changes over time, generating substances like oxides, compounds, and ketones due to factors such as storage environment, oxygen exposure, and light. As a solution for monitoring the freshness of edible fats and oils, gas sensors come into play. Escuderos et al. [73] conducted research on the feasibility of using quartz crystal microbalance (QCM) sensor arrays to differentiate the quality of olive oil samples. The experimental findings demonstrated the effective utilization of QCM sensor arrays for grading and monitoring the freshness of olive oil, offering the potential for the development of a low-cost, user-friendly, and rapid system to discern between virgin olive oil and extra-virgin olive oil.

### 4.6. Alcohol Products

The deterioration of alcohol products, such as wine, primarily arises from two key factors: the chemical composition of the beverage and the influence of microorganisms. Studies have identified oxidation as a significant contributing factor to the deterioration of alcoholic beverages [96]. When wine comes into contact with air, certain chemical components react with oxygen, resulting in alterations in taste, color, and texture. In addition to oxidation, other factors also impact the quality of wine products. External factors like light, temperature, and microorganisms can play crucial roles in affecting the overall quality and shelf life of wine.

Claudia et al. [75] conducted a study where they developed a low-cost gas sensor-based detection system to assess beer quality. Combining the gas sensor system with machine learning modeling, the researchers successfully predicted changes in beer aroma composition. The study utilized nine different gas sensors, namely ethanol, methane, carbon monoxide, hydrogen, ammonia/alcohol, benzene, hydrogen sulfide, ammonia, benzene/alcohol/ammonia, and carbon dioxide sensors. Artificial neural network models achieved strong correlations, with a coefficient of R = 0.97 for 17 volatile aromatic compounds (model 1) and R = 0.93 for sensory ratings (model 2). This study emphasizes the potential of integrating the developed gas sensor system with artificial neural network models as a cost-effective, rapid, dependable, and efficient approach for in-line beer quality assessment. In another study, Parvin et al. [97] employed electrochemical sensors utilizing cyclic voltammetry to monitor the oxidative degradation of wine. The study utilized a polyaniline (PANI) sensor, which demonstrated a negative shift in peak current and potential due to the formation of oxidation products like acetaldehyde. The PANI sensor proved effective in monitoring wine composition changes and offered advantages such as visual observation, affordability, high sensitivity, and selectivity.

## 5. Future Outlook

Despite the extensive research and application of gas sensors in food quality monitoring, certain challenges persist. Gas sensors demonstrate effectiveness in monitoring specific types of food or predetermined combinations of food products. However, the complex conditions, processes, and environments encountered during food storage and transportation present difficulties for gas sensors, leading to potential inaccuracies in food quality monitoring. Consequently, the limitations of gas sensor-based quality monitoring in detecting various food types and conditions restrict their ability to fully meet the requirements of food quality monitoring in the food industry.

In light of these challenges and limitations, this paper aims to summarize the development trends of gas sensor technology in food quality monitoring as follows:(a)Enhancing the immunity and stability of gas sensors holds paramount importance in the realm of food quality monitoring. The complexity of real-world food storage and transportation environments can introduce various challenges, potentially leading to inaccuracies in the data collected during freshness monitoring. Factors like sensor aging, reduced catalytic activity, and fluctuations in temperature and humidity can result in errors in sensor readings, adversely affecting the accuracy of detection. Therefore, focusing on enhancing the sensor array and the system’s resistance to interference becomes a crucial avenue for optimizing the identification accuracy of the system.(b)Advancements in gas sensors with cross-sensitivity to volatile compounds in food are crucial for effective food quality monitoring. To optimize the sensitivity, response time, cross-sensitivity, and doping process of these sensors, adjustments to their materials or structures become necessary. By incorporating such improvements, gas sensors can provide signals carrying more comprehensive information, necessitating the exploration of efficient data processing techniques. Techniques such as feature extraction and selection, pattern recognition, and regression modeling enable effective analysis and utilization of sensor signals, ultimately enhancing the accuracy and efficiency of food quality monitoring.(c)Developing highly sensitive gas sensors with improved detection limits remains a key focus in the field of sensor technology. Researchers are actively investigating materials with suitable surface and structural properties, exploring both rigid and flexible options to enhance sensor performance. By optimizing parameters such as sensitivity, response time, and cross-sensitivity, and refining doping processes and fabrication techniques, it is possible to achieve flexible gas sensors that offer remarkable sensitivity and stability. These sensors find significant applications in wearable electronic devices and electronic skin, presenting exciting possibilities for diverse industries. Despite these advancements, the selection of appropriate flexible substrates and sensing materials poses a primary challenge in the development of flexible gas sensors.(d)The miniaturization of sensors and real-time monitoring of smart data represent crucial directions in advancing gas sensors for food freshness monitoring. To effectively address the complexities of food storage and transportation, the expansion of test samples and the establishment of a comprehensive database characterizing the freshness of mixed food products are essential steps. Achieving more accurate food freshness monitoring requires not only expanding test samples but also optimizing model parameters and enhancing immunity to interference. As big data, the Internet of Things (IoT), 5G communication technology, and artificial intelligence continue to progress, future research should focus on achieving the portability of algorithmic models. This will enable faster and simpler food freshness recognition, promoting practical applications in various scenarios. Currently, freshness detection and recognition predominantly rely on odor information, offering relatively limited insights into food freshness using gas sensors. Therefore, future investigations could explore the integration of other detection methods, such as image recognition and colony detection, to achieve multi-sensory fusion detection and evaluation. This approach would lead to more accurate, intelligent, and rapid discrimination and monitoring of food freshness.

## 6. Conclusions

This paper presents a comprehensive review of gas sensor-based food quality monitoring from January 2011 to June 2023, with a specific focus on MOX gas sensors, colorimetric sensors, electrochemical sensors, and SAW gas sensors in various food quality applications, including freshness, ripeness, storage time, and hazardous ingredient detection. Despite the extensive research and application of gas sensors in food freshness monitoring, challenges persist, limiting their full potential in meeting the requirements of the food industry. MOX gas sensors, commonly used in intelligent detection systems, face sensitivity issues influenced by temperature, as well as long response and recovery times, hindering real-time online detection. SAW sensors, while exhibiting high sensitivity, suffer from weak anti-interference capabilities, and the modification of sensitive coatings can lead to prolonged response and recovery times. However, electrochemical sensors, with recent advancements incorporating nanomaterials, offer improved synthesis routes and beneficial properties. Colorimetric sensors, which are suitable for non-contact food detection, effectively display real-time food freshness status due to the gradual release of spoilage gases, with some offering visual representation. The continuous advancement of gas sensors is primarily driven by ongoing research and development efforts to explore and create new materials, innovative sensor structures, and improved manufacturing processes. Moreover, the emergence of sensor arrays paired with pattern recognition algorithms satisfies the most desired feature in sensor development, namely the plasticity of the sensor’s sensitivity.

This paper also addresses the development trends of gas sensor technology in food quality monitoring. The imperative lies in developing gas sensors with enhanced anti-interference capabilities, heightened stability, improved sensitivity, lower detection limits, and faster response times. A pressing need exists for gas sensors with cross-sensitivity toward a wide array of foods and food components. By leveraging advanced technologies like big data, IoT, 5G communication, and artificial intelligence, researchers can expand test samples, establish a comprehensive quality database encompassing mixed food products, and optimize model parameters based on these data. This endeavor will ultimately enable the intelligent monitoring and prediction of food quality.

## Figures and Tables

**Figure 1 foods-12-03966-f001:**
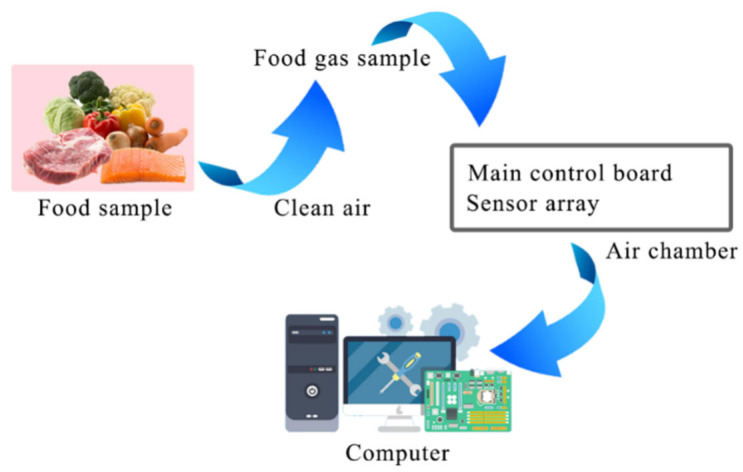
A flowchart of a gas sensor-based food gas detection system.

**Figure 2 foods-12-03966-f002:**
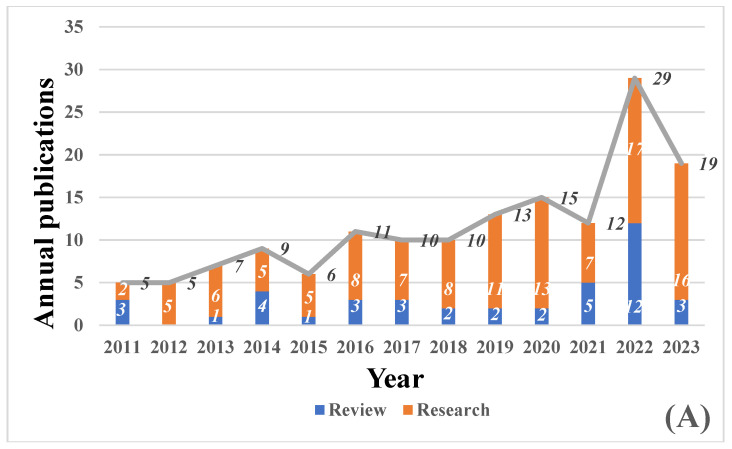
(**A**) Number and type of articles on gas sensor applications for food freshness monitoring from January 2011 to June 2023. Orange bars: research; dark blue bars: review. (**B**) Classes of gas sensors for applications in food freshness monitoring. Dark blue: metal oxide sensors; orange: surface acoustic wave gas sensors; gray: colorimetric sensors; yellow: electrochemical gas sensors; light blue: other types of gas sensors. (**C**) Food types for gas sensor-based freshness monitoring. Dark blue: metal oxide sensors; orange: surface acoustic wave gas sensors; gray: colorimetric sensors; yellow: electrochemical gas sensors; light blue: other types of gas sensors.

**Figure 3 foods-12-03966-f003:**
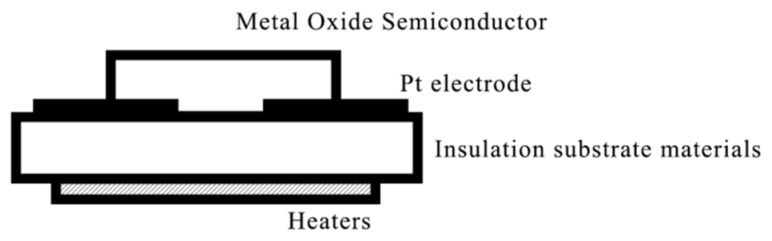
Schematic diagram of the structure of a metal oxide semiconductor gas sensor.

**Figure 4 foods-12-03966-f004:**
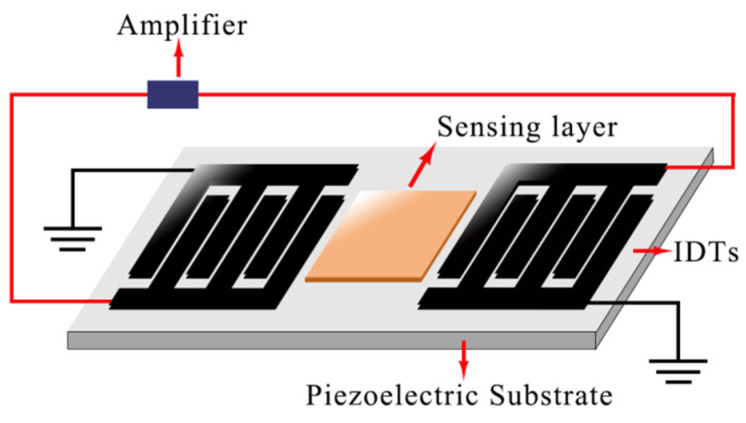
Schematic diagram of the structure of an SAW gas sensor. IDTs: interdigital transducers.

**Figure 5 foods-12-03966-f005:**
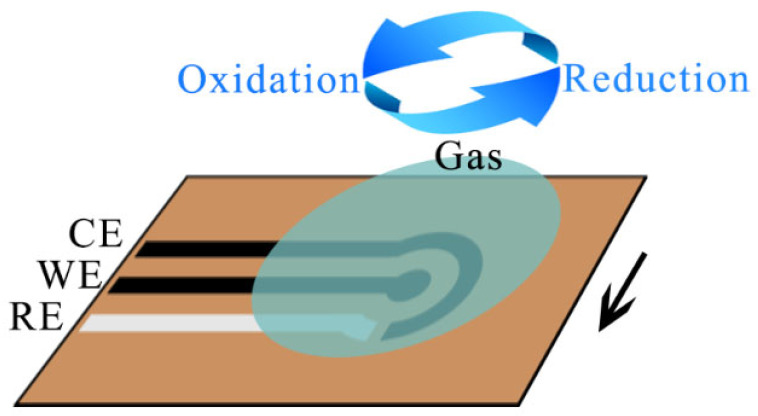
Schematic diagram of the structure of an electrochemical gas sensor. CE: counter electrode; WE: working electrode; RE: reference electrode.

**Table 1 foods-12-03966-t001:** Classification of food types for gas sensor-based freshness monitoring applications.

Type of Food	Class	Sensor Type	Gas	Reference
Aquatic Products	Pseudosciaena crocea	MOX sensor	Methane, ethane, dimethyl methane, ammonia, hydrogen sulphide, alcohol, toluene, xylene	[52]
	Large yellow croaker	MOX sensor	Sulfide, flammable gases, ammonia gas, ethanol, aromatic hydrocarbons, hydrocarbon component gas, methane, propane, butane, nitrogen oxides	[53]
	Oncorhynchus mykiss	MOX sensor	TVB-N	[20]
	Fish meal	MOX sensor	TVB-N	[19]
	Tilapia fish	Optical gas sensor	Short-chain alcohols and carbonyls, amines, sulphur compounds, aromatic, N-cyclic, acid compounds	[54]
	Fresh fish	Colorimetric sensor	Amines	[55]
	Grass carp fish, hairtail	Quartz crystal microbalance	Aldehydes	[56]
Meat products	Meat	MOX sensor	Ethanol, TMA	[57]
	Meat	Colorimetric sensor arrays	TMA	[35]
	Pork	Colorimetric sensor	CO_2_	[58]
	Chicken	MOX sensor	Volatile fatty acids	[21]
	Chicken	MOX sensor	NH_3_, TMA, ethanol, H_2_S	[59]
Fruit	Muskmelon	SAW sensor	Ethylene, carbon dioxide	[27]
	Apple, pear, kiwi fruit	Electrochemical sensor	Ethylene	[43]
	Banana	MOX sensor	Ethylene	[60]
	Apple	MOX sensor	Ethylene	[61]
	Rapeseed	MOX sensor	Volatile organic compounds	[62]
	Apple, banana, avocado, mango	Quartz crystal microbalance	Volatile organic compounds	[63]
Vegetables	Potato	MOX sensor, Electrochemical sensors	Carbon monoxide, ethylene oxide, nitric oxide.	[64]
	Potato	Colorimetric sensor	n-Hexadecanoic acid, pentadecanal, hexadecanoic acid, methyl ester, and methyl stearate	[65]
Dairy products	Milk	MOX sensor	Volatile organic compounds	[66]
	Cheese	MOX semiconductor gas sensor	Volatile organic compounds	[67]
	Cheese	MOX sensor	Volatile organic compounds	[18]
	Milk	Colorimetric sensor	Volatile organic compounds	[68]
Grain and oil products	Paddy, maize	MOX sensor	Ethanol, propane, butane, nitrogen oxides, carbon monoxide, hydrogen, hydrogen sulfide, methane	[69]
	Rice	MOX sensor	Aromatic compounds	[70]
	Rice	Colorimetric sensor	Volatile organic compounds	[71]
	Rice	Colorimetric sensor	Octaethylene glycol monododecyl ether, benzaldehyde	[72]
	Olive oil	Quartz crystal microbalance sensor array	Volatile organic compounds	[73]
Egg products	Egg	MOX sensor	Volatile organic compounds	[74]
Alcohol products	Beer	Sensor array	Aromatic compounds	[75]

## Data Availability

The data used to support the findings of this study can be made available by the corresponding author upon request.

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
