# Peer review of "Applications of Gas Sensing in Food Quality Detection: A Review"

_foods, 2023, doi:10.3390/foods12213966_

Round 1

Reviewer 1 Report

The main question addressed by the research is to investigate the application of gas sensors in food quality detection. The topic is interesting, but not so novel. The paper is well-organized and structured, and the text is clear and easy to read.

Literature search criteria (used words) should be defined.

Some references are not relevant (for example ref. 4). Some references are written in the wrong type (section 3.1. - (Han et al.,2014) is written instead of the number for the reference). References should be thoroughly revised.

Figures are very simple and do not give any novelty to the paper.

Double spaces have been find in many paragraphs.

The novelty is the main problem, there are a lot of papers (including reviews) with similar topics. It should be considered to perform a review in the last 5 years (not 13 as stated in the paper). Since previous papers were already reviewed in previous review papers.

Author Response

Thanks for your works.

Reviewer 2 Report

The manuscript attempts to provide a review of several types of gas sensors, focusing on applications in food quality assessment. While the text overall feels quite fluent, occasionally it is rather superficial, makes controversial statements or doesn't describe relevant details about the sensor operation.

- "A key characteristic of MOX sensors during operation is the carrier motion, governed by the semiconductor properties of the material. This results in MOX sensors exhibiting distinct responses to various target gases" It remains quite unclear how does the carrier motion yield distinct responses to various gases. The distinct responses have to have something to do with the distinct properties of the gas molecules. Similarly, in the next paragraph it is written "The resistance of these films changes in response to varying gas concentrations". But why? The physical principle of MOX semiconductor gas sensing has to be carefully rewritten.

- "However, recent advancements in research and development have led to the exploration of novel materials, including single metal oxide materials, composite metal oxide materials, and mixed metal oxide materials."  Quite some time (at least a decade) there has also been an intense research of 2D materials for resistive gas sensing (I'm not certain about their usage in food industry, though). Perhaps include in "Future Outlook".

- "Additionally, they are less susceptible to humidity, ensuring reliable performance in diverse environmental conditions" I'm not sure if this is so universally true. Perhaps more specific conditions (material, operating temperature, etc) with suitable references are necessary.

- From the current text it remains unclear what is the actual difference (in terms of the physical mechanism) between resistive and non-resistive semiconductor gas sensors.

- Food packaging (incl. vacuum packaging) is mentioned, but it probably remains unclear to most readers how is it possible to use electrical sensors (such as MOX sensors) to assess such sealed environment.

- In section 3.1 it is stated that resistive MOX sensors exhibit rapid response times, ensuring real-time monitoring capabilities. Yet, the conclusion section admits that MOX gas sensors face long response and recovery times, hindering real-time online detection. These are clearly opposing statements.

- Figures 4 and 5 contain undefined abbreviation IDT, CE, WE, RE.

- "the amperometric sensor measures the current as a function of the applied potential, while the potentiometric sensor measures the potential as a function of the current" The statement is probably incorrect. A standard definition says that amperometry monitors electric current while keeping a constant potential between the two electrodes, and potentiometry measures the electric potential while keeping a constant electric current.

- In the general description of the electrochemical sensors, it might be worth pointing out what are their main advantages, especially over the MOX sensors (which are also electrical)?

- "Gas sensor arrays offer additional benefits, including swift response rates" Again, the reader may be curious why would an array of sensors yield a more swift response than a single sensor?

- Figures 2 and 5 contain text elements possessing excessive font sizes.

There are possibilities to tighten the text. One can find very similar, repeating sentences, such as "Over the past decades, significant advancements have been achieved in the development and application of gas sensors" and "Over the past few decades, significant advancements have been made in gas sensor technology". As another example, the caption of Figure 2 enumerates the sensor types by color, but the information is already present on the Figure.

Author Response

Thanks for your works.

Reviewer 3 Report

The manuscript titled "A Review of Gas Sensor Applications in Food Quality Detection" authored by Minzhen Ma et al. aims to provide an overview of the various applications of gas sensors in monitoring food quality. While I anticipated a more in-depth exploration of the examples mentioned in the manuscript, I feel that even in its current state, it offers value to potential readers. Therefore, if the following comments are addressed.

1.      In the abstract: where you've referenced different types of gas sensors, including surface acoustic wave (SAW), colorimetric, and electrochemical sensors, it is recommended to exclude "Metal oxide" from the list since, unlike the others, it does not qualify as a distinct gas sensor method. All other types stand for a method but MOX is a class of materials. Therefore, you may consider using “Chemoresistive gas sensors” instead of MOX gas sensors.

2.      In the abstract it is mentioned that “Furthermore, the integration of data analytics and artificial intelligence into gas sensor arrays is discussed…” Indeed, I noticed that there were only a few fleeting references to data analysis and artificial intelligence in the context of gas sensors. However, the manuscript lacks an in-depth exploration of how these technologies have contributed to enhancing gas sensor performance and the specific advantages they offer in this regard.

3.      On Page 2: this sentence is not clear “As shown in Figure 1, the flowchart of the gas sensor-based food gas detection system is shown.” Please rewrite it.

4.      Regarding Figure 2C, could you please clarify the significance of the zeroes displayed in the bars? If they indicate the absence of data for a particular type of gas sensor, it might be advisable to omit them to prevent ambiguity.

5.      On page 5: Consider revising this sentence “ The fundamental functionality of a MOX sensor relies on reactions occurring when a gas interacts with a metal oxide or metal oxide semiconductor material adsorbed on its surface“ to the following: "The core functionality of a MOX sensor hinges on chemical reactions taking place when a gas interacts either directly with a metal oxide or with adsorbed species, such as oxygen ions or oxygen vacancies, on its surface"

6.      On page 6: “Additionally, they are less susceptible to humidity, ensuring reliable performance in diverse environmental conditions”. As an expert, I would not accept it. Indeed, unluckily, the majority of semiconductor-based resistive gas sensors are susceptible to humidity especially if they are going to work at neat room temperatures.

7.      On page 7: This sentence “Under typical circumstances, metal oxide sensors exhibit notably high resistance.” may not always hold true, as the resistance of a semiconductor can also be low, depending strongly on factors such as the type of semiconductor and its particle size.

8.      Figure 5 lacks clarity and could benefit from additional information. It would be helpful if you could elaborate on the terms "Oxidation O2" or "O2-Oxidation." Consider enhancing the figure with supplementary details or incorporating explanatory content into the manuscript to improve the overall understanding of this figure.

9.      In Table 1, Histamine is not a gas, and Reference 51 does not describe a gas sensor. Please remove it from the table. 

That is fine.

Author Response

Thanks for your works.

Reviewer 4 Report

This a comprehensive and well written review.

There are very very minor comments about the English, I Suggest the title be Applications of gas sensing in food quality detection

Figure 1 should be sensing

Table 1 should be NH3 with the 3 as a subscript

Reference 74 should be pH

 `I suggest potatoes should have coverage, as its a very important crop. Two papers are suggested: 

https://www.potatonewstoday.com/2022/08/24/sensor-that-can-smell-researchers-developed-new-biological-sensor-to-detect-soft-rot-in-potato-tubers/

Spencer-Phillips, P. T., Ratcliffe, N. M., Gunson, H. E., Ewen, R. J., De Lacy Costello, B. P., de Lacy Costello, B., …Spencer-Phillips, P. T. (2000). Development of a sensor system for the early detection of soft rot in stored potato tubers. Measurement Science and Technology, 11(12), 1685-1691. https://doi.org/10.1088/0957-0233/11/12/305

Reviewer 5 Report

This paper is a review paper to describe, as the authors say, the "gas sensors" to be applicable to foods. 

(1) The authors should show the definition of "gas sensor", because the difference is not clear from electronic noses, which are cited many times in the references. Or, these are the same? If differs. the authors should clarify it. 

(2) The authors refer to cyclic voltammetry (CV). To my knowledge, CV is not a gas sensor. Please explain it in more detail.  

(3) Do data in Fig. 2(A) include a part of e-noses or all of e-noses or none?  This comment is closely related to the above first.

(4) Figure 5 should be more clarified, because the flow of gas and the circumstance of CE, WE, and RE is not understood to readers. In addition, the authors should explain the role of CE and RE, by comparing it with the measurement method of amperometric and potentiometric sensors.

(5) The authors should explain the difference of "gas" and "odor" or "smell".

Round 2

Reviewer 1 Report

The paper can be accepted without any further changes.

Author Response

Thank you very much for your work and for your affirmation of our research.

Reviewer 2 Report

- In the revised manuscript, the phrase "electrochemical sensors
require regular maintenance and calibration to ensure accuracy and stability" occurs twice (and in the same paragraph).

- The sentence

Furthermore, the electrochemical sensors can
be categorized based on different transduction approaches, for example, "measurement of current as a function of the potential applied, known as amperometry sensor" and "measurement of potential as a function of the current applied, known as potentiometry sensor"

has been taken directly from ref. 41. Moreover, the sentence is probably wrong or misleading.  I would imagine that the measured electrical current (or electrical balance) changes in response to electrochemical reaction between the electroactive substance and the analyte gas. The authors should study the principle of electrochemical sensors more carefully and rephrase the sentence. Ref. 41 might be inappropriate in this context. A quick search revealed a more promising reference:

Baranwa et al, Electrochemical Sensors and Their Applications: A Review, Chemosensors 2022, vol. 10, 363.

The language is satisfactory.

Reviewer 5 Report

The authors insist that "An electronic nose primarily consists of a sensor array, a signal preprocessing module, and a pattern recognition engine. As a result, electronic noses differ significantly from gas sensors [4,5]." It is not true, which can be easily understood by considering the situation where "gas sensors" are used as sensors of "sensor" array of e-nose. In fact, many cited references in this paper refer to e-nose. One of the references [69] utilizes gas sensor in e-nose, and also references include the study to judge the quality, freshness, and spoilage using adequate software. From this point of view, the authors should reconsider the standing point to review either the sensing element or the system (i.e., e-nose). The authors should clarify whether they review gas sensors as sensing elements or review the system composed of gas sensors, which is nothing but e-nose.
